# Genetic Characterization and Pathogenicity of H7N7 and H7N9 Avian Influenza Viruses Isolated from South Korea

**DOI:** 10.3390/v13102057

**Published:** 2021-10-13

**Authors:** Eun-Jee Na, Young-Sik Kim, Yoon-Ji Kim, Jun-Soo Park, Jae-Ku Oem

**Affiliations:** Laboratory of Veterinary Infectious Diseases, College of Veterinary Medicine, Jeonbuk National University, Iksan 54596, Korea; ejna1212@naver.com (E.-J.N.); yoksik@naver.com (Y.-S.K.); kimyoonji102@naver.com (Y.-J.K.); spinyang@naver.com (J.-S.P.)

**Keywords:** avian influenza virus, H7N7, H7N9, wild bird

## Abstract

H7 low pathogenic avian influenza viruses (LPAIVs) can mutate into highly pathogenic avian influenza viruses (HPAIVs). In addition to avian species, H7 avian influenza viruses (AIVs) also infect humans. In this study, two AIVs, H7N9 (20X-20) and H7N7 (34X-2), isolated from the feces of wild birds in South Korea in 2021, were genetically analyzed. The HA cleavage site of the two H7 Korean viruses was confirmed to be ELPKGR/GLF, indicating they are LPAIVs. There were no amino acid substitutions at the receptor-binding site of the HA gene of two H7 Korean viruses compared to that of A/Anhui/1/2013 (H7N9), which prefer human receptors. In the phylogenetic tree analysis, the HA gene of the two H7 Korean viruses shared the highest nucleotide similarity with the Korean H7 subtype AIVs. In addition, the HA gene of the two H7 Korean viruses showed high nucleotide similarity to that of the A/Jiangsu/1/2018(H7N4) virus, which is a human influenza virus originating from avian influenza virus. Most internal genes (PB2, PB1, PA, NP, NA, M, and NS) of the two H7 Korean viruses belonged to the Eurasian lineage, except for the M gene of 34X-2. This result suggests that active reassortment occurred among AIVs. In pathogenicity studies of mice, the two H7 Korean viruses replicated in the lungs of mice. In addition, the body weight of mice infected with 34X-2 decreased 7 days post-infection (dpi) and inflammation was observed in the peribronchiolar and perivascular regions of the lungs of mice. These results suggest that mammals can be infected with the two H7 Korean AIVs. Our data showed that even low pathogenic H7 AIVs may infect mammals, including humans, as confirmed by the A/Jiangsu/1/2018(H7N4) virus. Therefore, continuous monitoring and pathogenicity assessment of AIVs, even of LPAIVs, are required.

## 1. Introduction

Avian influenza viruses (AIVs) are eight segmented, single-stranded negative-sense RNA viruses [1]. Based on viral surface proteins, hemagglutinin (HA) and neuraminidase (NA), influenza viruses are classified into 18 HA and 11 NA subtypes. H1-16 and N1-9 have been detected in avian species, but H17-18 and N10-11 have been discovered only in bats [2,3,4]. In particular, H5 and H7 are important subtypes because they have the potential to mutate into highly pathogenic avian influenza viruses (HPAIVs), causing severe clinical signs in poultry [5,6,7]. According to previous research, the H7 low pathogenic avian influenza virus (LPAIV) is a precursor to the H7 HPAIV [8,9,10,11,12,13,14,15].

H7 LPAIVs have been detected in poultry farms worldwide, and wild birds are regarded as the origin of H7 LPAIVs [16]. In wild birds, all nine neuraminidase (N1–N9) are found in H7 AIVs, and H7N7 viruses are the subtypes reported in the largest number of countries geographically [17]. In South Korea, two LPAIVs, H7N3 and H7N8, were detected in domestic ducks in 2007 [18]. Since then, through active national surveillance, H7 LPAIVs have been isolated continuously from poultry farms and wild birds, and H7N7 viruses are the most common subtypes in wild birds [19,20]. Viruses isolated from poultry are closely related to wild bird isolates [19]. Moreover, some H7 LPAIVs isolated from wild birds showed pathogenicity in chickens in laboratory experiments.

AIVs can cause cross-species transmission from birds to mammals [21]. Avian-like H7N7 viruses have been detected in equines and pinnipeds [22]. In addition, there are many cases of H7 AIV infection in humans. The first case of H7N7 LPAIV directly transmitted from avian to human was reported in 1996 [23]. The infected woman had contact with a duck; she suffered from conjunctivitis and recovered naturally. In 2003, 89 cases of H7N7 HPAIV human infections occurred in the Netherlands [24]. One of the 89 patients died of acute respiratory distress syndrome, and three were confirmed to have contracted the virus through transmission between humans. A recent human infection of H7N7 AIVs was observed in three poultry workers in Italy in 2013, and they showed only conjunctivitis but no respiratory syndromes [25].

In 2013, the first case of human infection with H7N9 AIVs was reported in Shanghai, and since then, the H7N9 virus has shown five epidemic patterns in China [26,27]. The H7N9 viruses that occurred in 2013 were low pathogenic viruses but gradually mutated, resulting in high pathogenic strains in 2017 [6,28]. By 2019, 1568 cases of H7N9 human infections were confirmed worldwide, of which 616 deaths occurred [29]. However, after the introduction of H7N9 vaccine in poultry in 2017 in China, human infection with H7N9 virus has decreased dramatically [30,31]. Since September 2019, no human infection has been reported according to the FAO H7N9 situation update. Moreover, human infections related to other subtypes of H7 LPAIVs, such as H7N2 and H7N3, have often been reported since the 2000s [32]. Severe human infection with H7N4 originating from backyard poultry was reported in Jiangsu in 2018 [33].

In 2021, two AIVs, H7N7 and H7N9, were isolated from wild birds in South Korea. The molecular and phylogenetic characterizations of the two H7 Korean AIVs were analyzed in this study. In addition, pathogenicity in mammals was evaluated using a murine model.

## 2. Materials and Methods

### 2.1. Sampling

Fresh fecal samples (2499) of wild birds were collected between October 2020 and March 2021 by active monitoring of wild bird habitats. The collected samples were immediately shipped to the laboratory and stored between 2 and 8 °C until further analysis.

### 2.2. Virus Isolation

Each sample was suspended in phosphate-buffered saline (PBS, pH 7.4) supplemented with 100 U/μL of penicillin and 100 U/μL of streptomycin and centrifuged at 2800× *g* for 10 min. Supernatants were filtered with a 0.45-μm pore size syringe filter (GVS, Sanford, ME, USA). The filtered samples were inoculated into the allantoic cavities of 9–11-day-old specific-pathogen-free embryonated chicken eggs (Seong-Min Inc., Hwaseong, Korea) and incubated at 37 °C for 72 h. After chilling at 4 °C, the allantoic fluids were harvested and checked for AIV using hemagglutinin assay (HA assay) according to the World Organization for Animal Health (OIE) standard [34]. The AIV was verified in the HA-positive samples using the M gene quantitative real-time reverse transcription-polymerase chain reaction (qRT-PCR) kit (iNtRON Biotechnology, Seongnam, Korea).

### 2.3. Subtyping and Species Identification

Viral RNA was extracted using a Miracle-AutoXT Automated Nucleic Acid Extraction System (iNtRON Biotechnology, Seongnam, Korea). To identify subtypes, qRT-PCR was conducted using the TOPreal™ One-step RT qPCR Kit (Enzynomics, Daejeon, Korea) with influenza-specific primers as previously described [35]. Species identification was performed using mitochondrial cytochrome C oxidase I mitochondrial gene in the DNA of fecal samples, as previously described [36].

### 2.4. Sequencing and Phylogenetic Analysis

Extracted viral RNAs were reverse transcribed to complementary DNAs (cDNAs) using a cDNA Synthesis kit (Wizbiosolutions, Seongnam, Korea) according to the manufacturer’s instructions. The synthesized cDNA was amplified using PCR and universal primers [37]. The amplicons were sequenced using an ABI 3730xl DNA analyzer (Applied Biosystems, Foster City, CA, USA). Nucleotide sequences were aligned using the ClustalW algorithm in BioEdit v.7.0.9.0. Phylogenetic trees were constructed using the maximum-likelihood (ML) method with 1000 bootstrap replications in MEGA (version 7.0) software [38]. The best substitution model for each segment was estimated using jModelTest (version 2.1.10) software [39,40]. The best models were the general time reverse model with invariable sites (GTR + I) for HA; GTR with gamma distribution (GTR + G) for PB2, PA, and NP; GTR + G + I for PB1 and M; Hasegawa Kishino Yano model with invariable sites (HKY + I) for NA7; and HKY + G for NA9 and NS. The reference sequences were retrieved from the National Center for Biotechnology Information Influenza Virus Resource (http://www.ncbi.nlm.nih.gov/genomes/FLU/FLU.html, accessed on 15 March 2021) and the Global Initiative on Sharing Avian Influenza Data (GISAID; http://www.gisaid.org, accessed on 15 March 2021). The nucleotide sequences of two H7 Korean AIVs were compared with sequences of NCBI-Influenza Virus Resource and GISAID database and reference sequences with high nucleotide identity were downloaded. 

Maximum clade credibility (MCC) trees of the HA gene were constructed using Bayesian Evolution Analysis Sampling Trees (BEAST) version 1.10.4, to estimate the time to the most recent common ancestor [41]. We constructed a phylogenetic tree using H7 gene sequences obtained from GISAID between 2016 and 2021 in Asia. We used the HKY nucleotide substitution model with a gamma distribution and uncorrelated lognormal distribution relaxed clock method. All chains were run in 100,000,000 generations and the effective sample size values were greater than 200 in Tracer 1.7.1 (http://tree.bio.ed.ac.uk/software/tracer/, accessed on 27 April 2021). The MCC tree was constructed using TreeAnnotator v1.8.1 (http://beast.bio.ed.ac.uk/TreeAnnotator/, accessed on 6 May 2021) with 10% burn-in cutoffs and visualized with FigTree 1.4.4 (http://tree.bio.ed.ac.uk/software/figtree/, accessed on 6 May 2021).

### 2.5. Pathogenicity in Mice

Forty female 6-week-old BALB/c mice (Samtaco, Osan, Korea) were used in this study. The mice were randomly divided into three groups, with 15 mice in the two experimental groups and 10 mice in the control group. After being anesthetized with avertin (Sigma-Aldrich, Milan, Italy), each group of 15 mice was inoculated intranasally with 10^7^ egg infectivity dose (EID_50_) in a volume of 50 μL of 20X-20 (H7N9) and 34X-2 (H7N7) strains. Ten BALB/c mice were inoculated intranasally with 50 μL of PBS as the negative control group. The mice were monitored daily for body weight, body temperature, and clinical signs for 14 days post-infection (dpi). Three mice from the experimental group and two from the control group were sacrificed at 1, 3, 5, 7, and 14 dpi. The right lungs of mice were collected and were analyzed for virus detection using the M gene qRT-PCR kit (iNtRON Biotechnology, Seongnam, Korea), and the positive lung tissue as detected by qRT-PCR was used for virus titration using EID_50_ via the Reed and Muench method [42]. The left lobe of the mice was collected for pathology study and fixed in formalin for hematoxylin and eosin staining. The paraffin-embedded lung tissue was sectioned at 3 μm. For seroconversion and antigenic analysis, serum samples were collected from three mice in the experimental group and two mice from the control group at 14 dpi. The collected serum was treated with a receptor-destroying enzyme (Denka Seiken, Tokyo, Japan) and incubated for 18 h in a 37 °C water bath. Serum was used for hemagglutination inhibition (HI) tests following the OIE standard [34].

### 2.6. Statistical Analysis

The data and graphs shown in this study were analyzed using analysis of variance (ANOVA) in GraphPad Prism (version 6.0; GraphPad Software Inc., San Diego, CA, USA). The values shown are the mean with standard deviation (SD) or standard error of the mean (SEM). Statistical significance was set at *p* < 0.05.

### 2.7. Ethical Approval

All experiments were conducted at the animal biosafety level 2+ (ABSL + 2) facility. This study was approved by the Jeonbuk National University Laboratory Animal Research Center (JBNU 2021-0129).

## 3. Results

### 3.1. Identification of H7NX AIV Isolates

In January 2021, H7N9 and H7N7 AIVs were isolated from JeonJucheon (35°52′9.00′′ N, 127°6′20.28′′ E) and Dongjingang (35°41′22.53′′ N, 126°50′50.19′′ E), respectively. Cytochrome C oxidase I gene sequencing revealed that fecal samples came from Anas platyrhynchos (mallards). The H7N9 and N7N7 AIVs were designated as A/mallard/South Korea/JB20X-20/2021(H7N9; abbreviated as 20X-20) and A/mallard/South Korea/JB34X-2/2021(H7N7; abbreviated as 34X-2), respectively.

### 3.2. Molecular Characterization of H7 AIV Isolates

The whole genomes of 20X-20 and 34X-2 viruses were obtained and deposited in GenBank with the accession numbers MZ803114-MZ803121 and MZ803125-MZ803132, respectively. The obtained full gene sequences of 20X-20 and 34X-2 were compared with four selected reference H7 viruses: A/chicken/Jiangsu/1/2018 (H7N4; chicken/1), A/Jiangsu/1/2018 (H7N4; Jiangsu/1), A/Anhui/1/2013 (H7N9; Anhui/1), and A/Italy/3/2013 (H7N7; Italy/3) (Table 1) (14, 15, 19). The Jiangsu/1, Anhui/1, and Italy/3 viruses originate from the H7 AIV and cause human infections. The chicken/1 virus was isolated from chickens in the backyard of humans infected with the Jiangsu/1 virus. The amino acid sequences of the HA cleavage sites of 20X-20 and 34X-2 were ELPKGR/GLF, indicating they are LPAIVs. The amino acids at positions 186, 190, 225, 226, 227, and 228 of the HA gene (numbering based on H3) are related to host receptor binding efficiency [43,44,45]. Except for 186V and 226L in Anhui/1, all H7 viruses had 186G, 190E, 225G, 226Q, 227S, and 228G, indicating that they prefer avian receptors over human receptors. NA stalk deletions (69–73) were not detected in any of the H7 viruses except for the Anhui/1 virus. However, drug resistance-associated mutations were identified at the 117 position in the NA genes of 20X-20 and 34X-2.

The amino acid mutations of the two H7 Korean AIVs were mostly similar, except for K142R in the PA gene of 20X-20 and M105V in the NP gene of 34X-2. The PA gene of the two H7 Korean AIVs was clustered in the same subgroup in the phylogenetic tree but showed a different amino acid mutation at the 142 position. The amino acid substitutions of E627K in the PB2 gene, known to increase virulence in mice and human adaptation markers, were observed in Jiangsu/1 but not in the two H7 Korean AIVs [46,47]. Additionally, the amino acid substitutions of I292V of the PB2 gene, which increase the polymerase activity in both and avian cells were not identified in the two H7 Korean AIVs. The amino acid substitutions of N66S in the PB1-F2 gene, which increased virulence in mice, were observed in the two H7 Korean AIVs compared to the chicken/1, which most recently caused human infections [48]. The other mutations related to the virulence of the virus and drug resistance are shown in Table 1.

### 3.3. Phylogenetic Analysis of H7 AIVs Isolates

An ML tree analysis of the HA gene showed that the HA genes of 20X-20 and 34X-2 belonged to the Eurasian lineage (Figure 1). The HA genes of 20X-20 and 34X-2 showed 98.5% nucleotide identities with each other and were most closely related to A/wild duck/South Korea/KNU18-106/2018(H7N7), with identities of 97.8% and 98.5%, respectively (Table 2). Notably, the two H7 Korean AIVs showed high nucleotide similarity to the A/Jiangsu/1/2018(H7N4) virus (97.5% and 97%, respectively), which are human influenza viruses originating from AIVs. However, the HA genes of 20X-20 and 34X-2 viruses showed low nucleotide similarity to the HA gene of A/Italy/3/2013(H7N7) (90.2% each) and A/Anhui/1/2013(H7N9) (88.6% and 88.9%, respectively), which cause human infections.

To estimate the times of origins of the HA gene of the two H7 Korean AIVs, we reconstructed the MCC tree. As the HA gene of the two H7 Korean AIVs belonged to the Asian lineage, we reconstructed the MCC tree using the H7 gene of LPAIVs isolated between 2016 and 2021 in Asia. In the MCC tree analysis, HA genes formed two distinct genetic subgroups (G1 and G2) (Figure 2). The H7 LPAIV in the G1 subgroup was identified as the one circulating in South Korea in 2017, and the H7 LPAIVs in the G2 subgroup were closely related to the H7 LPAIV isolated in China between 2016 and 2017. The HA genes of 20X-20 and 34X-2 belonged to the G1 subgroup.

The results of the phylogenetic tree of the NA gene showed that the NA gene of the two H7 Korean AIVs belonged to the Eurasian lineage (Appendix A). The NA genes of 20X-20 and 34X-2 showed the highest nucleotide identities with the NA gene of A/Anas platyrhynchos/South Korea/JB31/96/2019(H11N9) (98.2%) and A/mallard/Korea/A15/2016(H7N7) (97.4%), respectively. However, the NA gene of the two H7 Korean AIVs clustered distinct subgroups with human influenza viruses.

Most of the internal genes (PB2, PB1, PA, NP, M, and NS) of the two H7 Korean AIVs belonged to the Eurasian lineage, except for the M gene of 34X-2 (Appendix A). The PB2, PB1, PA, NP, M, and NS genes of 20X-20 and 34X-2 shared 95.1%. 95%, 96.7%, 94.1%, 93.9%, and 99.5% nucleotide identities, respectively. The NP and M genes, which showed less than 95% nucleotide similarity, were clustered into different subgroups. The PA gene of 20X-20 was clustered into the same subgroup as the PA gene of A/Mandarin duck/Korea/WB246/2016(H5N6), which is an internal gene of HPAIV.

### 3.4. Pathogenicity in Mice

To determine the pathogenicity of 20X-20 and 34X-2 viruses in a murine model, 6-week-old BALB/c mice were intranasally inoculated with each virus. All mice were monitored for 14 days post-infection (dpi). Although mice infected with 20X-20 showed no clinical symptoms, mice infected with the 34X-2 showed ruffle fur and significantly lower body weight than mice infected with 20X-20 and PBS on 7 dpi (*p* < 0.01) (Figure 3). The mortality rate was 0% in all mice.

To observe the viral load in mouse lungs, the mice were sacrificed on 1, 3, 5, 7, and 14 dpi. Until 7 dpi, the virus antigens were detected in the lungs of mice, but there was no viral detection in the lungs of all groups on 14 dpi (Table 3). Further, 20X-20 replicated in the lungs with the highest titer on 5 dpi (2.80 ± 2.43 EID_50_/mL), which decreased dramatically on 7 dpi (0.48 ± 0.83 EID_50_/mL). However, 34X-2 showed a higher titer than 20X-20 in the lungs on 1, 3, 5, and 7 dpi. Moreover, 34X-2 replicated in the lung with the highest titer on 3 dpi (4.28 ± 0.14 EID_50_/mL), which decreased slightly until 7 dpi (3.86 ± 0.80 EID_50_/mL) but maintained relatively high viral titer.

Histopathological analysis of the lung tissues showed different degrees of inflammation between the groups challenged with 20X-20 and 34X-2 viruses. The inflammatory cells infiltrated into the peribronchiolar and perivascular regions of the lungs of mice infected with 20X-20 (5 and 7 dpi) and 34X-2 (3 and 7 dpi) are shown in Figure 4. Severe interstitial pneumonia and degeneration of the bronchial epithelium were observed in the lungs of mice infected with 34X-2 on 7 dpi.

Seroconversion and antigenic analyses were performed using mouse serum on 14 dpi. Seroconversion was observed in all mice and two of three mice challenged with 20X-20 and 34X-2, respectively (Table 4). The antigenic result of the cross-HI test revealed no difference in antigenic patterns between the two H7 Korean viruses. The 20X-20 virus showed identical HI titers against its homologous antiserum (8, 32, 32) and 34X-2 antiserum (8, 32, 32). The 34X-2 virus also showed identical HI titers against its homologous antiserum (8, 32) and 20X-2 antiserum (8, 32).

## 4. Discussion

LPAIVs cause mild respiratory syndromes and decreased feed intake and egg production in poultry, leading to significant economic losses [73,74]. Although H7 HPAIVs are yet to cause an outbreak in South Korea, H7 LPAIVs have caused an outbreak, resulting in economic losses. In 2010 and 2011, H7N2, H7N6, H7N7, and H7N8 were isolated from domestic ducks and over 197,000 poultry were culled to eradicate the disease [17]. Moreover, H7 LPAIVs can infect humans and cause pathogenicity [75]. Therefore, in this study, we analyzed the genetic characterization and pathogenicity of two LPAIV viruses, H7N7 and H7N9, isolated from wild birds in South Korea in 2021.

We isolated H7N9 (20X-20) and H7N7 (34X-2) AIVs through active influenza surveillance from October 2020 to March 2021. Both viruses have ELPKGR/GLF at the HA cleavage site, indicating they are LPAIVs, and the same cleavage site has been reported in human influenza virus [33]. In particular, the receptor binding sites involved in interspecies transmission are located in the HA gene [76]. There were no amino acid substitutions of G186V, E190D, Q/G225D, Q226L, S227N, and G228S in the HA gene with known effect on receptor binding to mammalian tissue, demonstrating that the two Korean H7 AIVs would likely have an avian receptor binding specificity. However, since the characterization was analyzed only at the sequence level, an actual receptor binding experiment would need to be performed to demonstrate avian receptor binding specificity.

The amino acid substitutions of E627K and D701N in PB2, which is a well-known mammalian virulence factor, were not observed in the two H7 Korean AIVs. However, several amino acid substitutions related to enhanced virulence in mice were identified in the two H7 Korean AIVs: L89V, G309D, and T339R in the PB2 gene; D622G in the PB1 gene; N66S in the PB1-F2 gene; V286A and M437T in the NP gene; N30D, I43M, and T215A in the M1 gene; and P42S, L103F, and C138F in the NS1 gene. Although the two H7 Korean AIVs have avian receptor binding preference in the HA gene, they showed pathogenicity in the murine model. This result suggests that the presence and combination of other amino acid mutations may affect virulence in mice. Therefore, continuous studies are needed on how amino acid mutations play an important role in LPAIVs replication in the mammalian cell. The two H7 Korean AIVs carried I117T amino acid mutations in the NA gene, which reduced the antiviral drug susceptibility. A/Jiangsu/1/2018 (H7N4; Jiangsu/1), which is closely related to chicken/1, was susceptible to antivirals (oseltamivir) and they had no I117T mutations in the NA gene [66]. This result suggests that the two H7 Korean AIVs may be resistant to antiviral drugs, but further studies are required to confirm this finding. 

The eight segments of the two H7 Korean AIVs belonged to the Eurasian lineage. The MCMC tree of the HA gene revealed that the HA gene of the two H7 Korean AIVs clustered in the G1 group, suggesting that the HA gene of these two strains likely originated from the AIV circulating in Korea in 2017. Moreover, the PB2, PB1, PA, HA, and NA genes of 20X-20 and the PB2, PB1, HA, and NA genes of 34X-2 showed high nucleotide similarity to AIVs of Korean origin. The NP, M, and NS genes of 20X-20 and the PA, NP, and NS genes of 34X-2 showed high nucleotide identity with Asian AIV strains, suggesting that the segments were introduced to Korea by intracontinental reassortment. However, the M gene of the 34X-2 virus belonged to the American lineage, and this result demonstrates intercontinental reassortment of AIVs.

In the ML tree analysis of the HA gene, two H7 Korean AIVs showed low nucleotide similarity to the HA gene of A/Italy/3/2013(H7N7) and A/Anhui/1/2013(H7N9), which originate from AIVs and cause human infections. The ML tree analysis of the NA gene of the two H7 Korean AIVs showed the same results as the HA gene. However, the HA gene of the two H7 Korean AIVs showed a high similarity to the A/Jiangsu/1/2018(H7N4) virus, which recently caused a severe human infection in China [33]. China is adjacent to South Korea, and there is a possibility of novel viruses being introduced into South Korea. Therefore, continuous surveillance of AIVs is required.

Previous studies have shown that H7 LPAIVs can replicate in the lungs of murine models [77,78,79]. The two H7 Korean AIVs were isolated from the lungs of infected mice without prior host adaptation. All infected mice showed a 0% mortality rate. However, mice infected with 34X-2 showed body weight loss and clinical symptoms of ruffled fur on 7 dpi. In the histopathology analysis, inflammation in the peribronchiolar and perivascular regions was observed in both lungs of mice infected with the two H7 Korean AIVs. Inflammation was most severe in 34X-2-infected mice on 7 dpi. Further, 20X-20 and 34X-2 showed the highest viral titer of 2.80 log_10_EID_50_/mL on 5 dpi and 4.28 log_10_EID_50_/mL on 3 dpi, respectively. However, the two H7 Korean AIVs were not detected in the lungs of mice on 14 dpi. Moreover, the two H7 Korean AIVs indicated a low seropositive rate on 14 dpi, and even one of three mice infected with 34X-2 viruses showed no seroconversion. This result suggests that the two H7 Korean AIVs can replicate in mice and that the viruses can recover naturally. However, the lungs of the mice showed severe histopathological lesions, suggesting that AIVs can also show pathogenicity in other mammals. The antigenic analysis using the cross-HI test demonstrated that there were no significant antigenic differences between 20X-20 and 34X-2. However, 20X-20 and 34X-2 showed differences in pathogenicity in the murine model. AIVs can increase virulence by reassortment with other AIVs and the introduction of another segment may facilitate transmission of AIVs [80,81]. This result suggests that internal genes of 34X-2, especially the M, NP, and NA genes, which clustered into distinct subgroups with 20X-20, may enhance the pathogenicity of 34X-2. 

Our study provides information about genetic characterization and pathogenicity in murine models of H7H7 and H7H9 AIVs isolated from wild birds in South Korea in 2021. AIVs have been a concern because of the potential risks of human infection. Human infection with H7N9 AIVs has been reported to be more prevalent than other subtypes [29]. Although the H7N7 and H7N9 AIVs are LPAIVs, they exhibited pathogenicity in a murine model. Our findings suggest that the two H7 Korean AIVs have the potential to infect mammals, including humans. Moreover, H7 LPAIVs have the potential to mutate to HPAIV. Therefore, continuous surveillance of H7 AIVs is required.

## Figures and Tables

**Figure 1 viruses-13-02057-f001:**
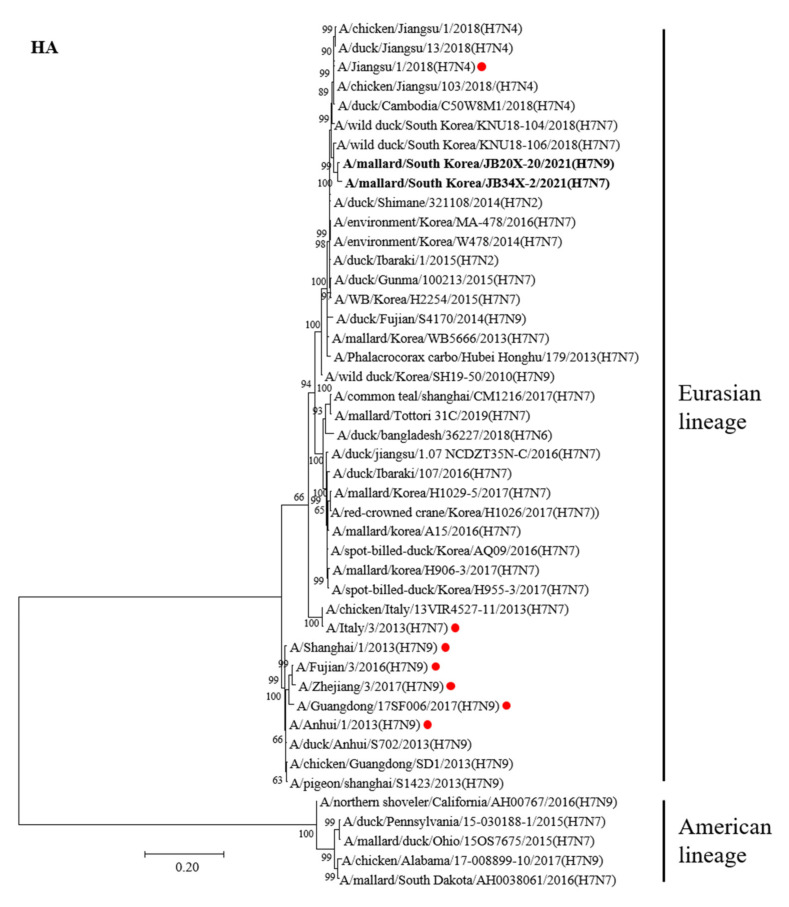
Phylogenetic tree of the HA gene of the H7 avian influenza virus (AIV) nucleotides. The tree was analyzed using the maximum-likelihood (ML) method with 1000 bootstrap replication and only bootstrap values more than 50% are shown. The two H7 Korea AIVs isolated in this study are shown in bold. The human influenza viruses are indicated by red circles.

**Figure 2 viruses-13-02057-f002:**
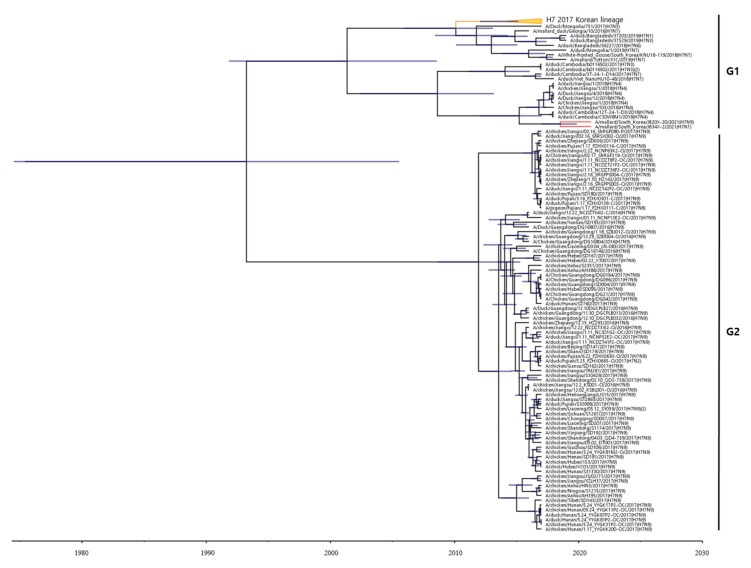
Maximum clade credibility (MCC) trees of the HA gene of H7 LPAIVs isolated between 2016 and 2021 in Asia. The MCC tree were constructed using the uncorrelated lognormal distribution relaxed clock method in BEAST v1.10.4. The ESS values were greater than 200. Posterior probabilities >0.8 are provided in the tree. The horizontal axis indicates the time scale, and the unit is 10 years. The two H7 Korean AIVs are shown in red. The H7 LPAIVs circulating in South Korea in 2017 are shown in orange.

**Figure 3 viruses-13-02057-f003:**
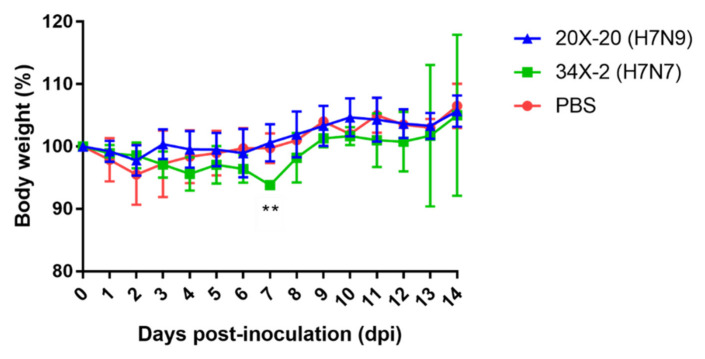
Body weight changes in mice infected with 20X-20 and 34X-2 viruses. BALB/c were intranasally inoculated with 10^7^EID_50_ of each virus. The mice were observed for 14 days. The data indicate the mean and standard error of the mean (SEM). The data were analyzed using two-way ANOVA with Tukey’s multiple comparisons test. The asterisks show that the body weight of 34X-2 is significantly different from that of 20X-20 and the PBS group (** *p* < 0.01).

**Figure 4 viruses-13-02057-f004:**
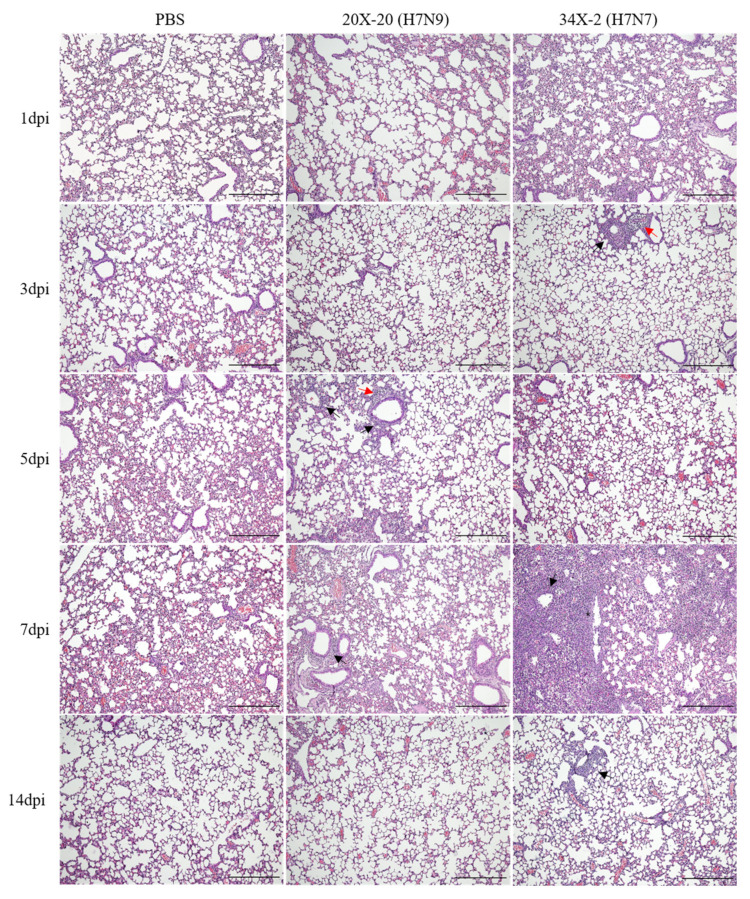
Histopathology analysis of the lungs of mice inoculated with 20X-20 and 34X-2. Lung tissue sections from mice were stained with hematoxylin and eosin. Inflammatory cells infiltrated into the peribronchiolar (black arrow) and perivascular regions (red arrow) of the lungs of mice infected with 20X-20 (H7N9) and 34X-2 (H7N7) viruses can be visualized. Interstitial pneumonia and degeneration of bronchial epithelium (asterisk) were observed in 34X-2 (H7N7)-infected mice on 7 dpi. The original magnification is ×100 and the scale bar indicates 200 µm. Dpi, days post-infection.

**Table 1 viruses-13-02057-t001:** Molecular characterization of H7 avian influenza virus (AIV) isolates.

Viral Protein	Amino Acid Residue	Virus Strains						Comments	Reference
		20X-20 (2021)	34X-2 (2021)	Chicken/1 (2018)	Jiangsu/1 (2018)	Anhui/1 (2013)	Italy/3 (2013)		
HA	Cleavage site	ELPKGR/GLF	ELPKGR/GLF	ELPKGR/GLF	ELPKGR/GLF	ELPKGR/GLF	ETPKRRERR/GLF	LPAIV -monobasic	[49]
	G186V	G	G	G	G	V	G	Increased α2-6 binding	[45]
	E190D	E	E	E	E	E	E		[43]
	Q/G225D	G	G	G	G	G	G		[43]
	Q226L	Q	Q	Q	Q	L	Q		[43]
	S227N	S	S	S	S	S	S		[44]
	G228S	G	G	G	G	G	G		[43]
PB2	L89V	V	V	V	V	V	V	Increased polymerase activity in mammalian cell lines and mice	[46]
	I147T	I	I	I	I	I	T		[50]
	I292V	I	I	I	I	V	I		[51]
	G309D	D	D	D	D	D	D		[46]
	T339K	K	K	K	K	K	K		[46]
	K389R	R	R	R	R	K	R		[52]
	E627K	E	E	E	K	K	E	Increased virulence in mice	[53]
	D701N	D	D	D	D	D	D		[54,55]
	V598T/I	T	T	T	T	V	T	Increased polymerase activity in mammalian cell line	[52]
PB1	C38Y	Y	Y	Y	Y	Y	Y	Increased polymerase activity in mammalian cell line and pathogenicity in chicken	[56]
	D622G	G	G	G	G	G	G	Increased polymerase activity and virulence in mice	[57]
PB1-F2	N66S	S	S	N	N	N	N	Increased virulence in mice	[48]
PA	S37A	A	A	A	A	S	A	Increased polymerase activity in mammalian cell line	[58]
	K142R	R	K	K	K	K	K		[55]
	N383D	D	D	D	D	D	D	Increased pathogenicity in ducks	[59]
	N409S	S	S	S	S	N	S	Increased polymerase activity in mammalian cell line	[58]
NP	I41V	I	I	I	I	I	I	Increased polymerase activity in mammalian cell line	[60]
	M105V	M	V	M	M	V	V	Increased virulence in chicken	[61]
	A184K	K	K	K	K	K	K		[62]
	F253I	I	I	I	I	I	I	Increased virulence in mice	[63]
	V286A	A	A	A	A	A	A		[64]
	M437T	T	T	T	T	T	T		
NA	69–73 deletion (QISNT)	No	No	No	No	Yes	No	Increased virulence in mice	[65]
	I117T	T	T	I	I	T	T	Increased resistance to antiviral drugs (oseltamivir and zanamivir)	[66]
M1	N30D	D	D	D	D	D	D	Increased virulence in mice	[50]
	I43M	M	M	M	M	M	M		[67]
	T215A	A	A	A	A	A	A		[50]
M2	L26F	L	L	L	L	L	L	Increased resistance to antiviral drugs (amantadine and rimantadine)	[68]
	S31N	S	S	S	S	N	S		[68]
NS1	P42S	S	S	S	S	S	S	Increased virulence in mice	[69]
	D92E	D	D	D	D	D	D	Increased virulence in mice	[70]
	L103F	F	F	F	F	L	F	Increased replication and virulence in mice	[71]
	C138F	F	F	F	F	F	F	Increased replication in mammalian cell line	[72]

**Table 2 viruses-13-02057-t002:** Sequence identities of the A/mallard/South Korea/JB20X-20/2021 (20X-20) and A/mallard/South Korea/JB34X-2/2021 (34X-2) genome.

Gene	20X-20	Genetic Identity	34X-2	Genetic Identity
PB2	A/wild_duck/South_Korea/KNU18-106/2018(H7N7)	98.3%	A/red-crowned crane/South Korea/H1026/2017(H7N7)	96.6%
PB1	A/red-crowned crane/South Korea/H1026/2017(H7N7)	95%	A/wild_duck/South_Korea/KNU18-104/2018(H7N7)	93.6%
PA	A/wild_duck/South_Korea/KNU18-106/2018(H7N7)	99.1%	A/wild_bird/Eastern_China/1758/2017(H5N3)	98.6%
HA	A/wild_duck/South_Korea/KNU18-106/2018(H7N7)	97.8%	A/wild_duck/South_Korea/KNU18-106/2018(H7N7)	97.3%
NP	A/common teal/Shanghai/NH110923/2019(H1N1)	98.8%	A/duck/Mongolia/926/2019(H5N3)	99.3%
NA	A/Anas platyrhynchos/South Korea/JB31-96/2019(H11N9)	98.2%	A/mallard/Korea/A15/2016(H7N7)	97.4%
MP	A/duck/Mongolia/916/2018(H3N8)	99.7%	A/northern pintail/Alaska/362/2013(H3N8)	99.4%
NS	A/duck/Bangladesh/37509/2019(H8N4)	99.6%	A/duck/Bangladesh/37509/2019(H8N4)	99.6%

**Table 3 viruses-13-02057-t003:** Replication of H7 isolates in the lungs of 6-week-old BALB/c mouse. Values shown are the number of infected mice per number of inoculated mice. Values in parentheses are viral titers (log_10_EID_50_/mL). The viral titers are expressed as the mean ± standard deviation (SD) of the samples. The viral titers from the lungs of mice were calculated using the Reed and Muench method.

Isolate	Days Post Infection (Mean ± SD, log_10_EID_50_/mL)				
	1	3	5	7	14
20X-20 (H7N9)	3/3 (2.36 ± 1.01)	3/3 (1.31 ± 2.28)	3/3 (2.80 ± 2.43)	3/3 (0.48 ± 0.83)	0/3
34X-2 (H7N7)	3/3 (2.71 ± 2.36)	3/3 (4.28 ± 0.14)	3/3 (4.14 ± 0.10)	3/3 (3.86 ± 0.80)	0/3

Dpi, days post-infection; EID_50_, 50% Egg-infective dose.

**Table 4 viruses-13-02057-t004:** Seroconversion in 6-week-old BALB/c mouse infected with H7 isolates and antigenic analysis of H7 isolates. Serum was collected on 14 dpi and treated for 18 h with a receptor-destroying enzyme. Seroconversion was confirmed using the hemagglutination inhibition (HI) assay.

		Seroconversion: Positive/Total (HI Titers)	
	SerumVirus	20X-20	34X-2
Antigen			
20X-20		3/3 (8, 32, 32)	2/3 (8, 32)
34X-2		3/3 (8, 32, 32)	2/3 (8, 32)

## Data Availability

Not applicable.

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
