# Peer review of "Genetic Characterization and Pathogenicity of H7N7 and H7N9 Avian Influenza Viruses Isolated from South Korea"

_viruses, 2021, doi:10.3390/v13102057_

Round 1
Reviewer 1 Report
The authors isolated two H7 viruses and analyzed the genetic characters and evaluated the pathogenicity in the mammalian model. The results indicated that the H7 viruses are actively reassorting in nature, which highlights the potential risk to poultry production and human health. The manuscript needs to be extensively revised before it can be considered as a publication.
Specific comments:
- Page 1. Abstract. The authors described that “There were no amino acid substitutions at the receptor-binding site of the HA”. The authors should indicate a reference strain that they compared with in their analysis. Without a reference virus, it’s difficult to understand the so-called substitutions.
- Page 1. Introduction. Several important papers (PMID: 29151586 and PMID: 7483266) reporting the findings that the H5 and H7 viruses have the potential to mutate from LPAIV to HPAIV should be cited in the area where the reference 3 is cited.
- The authors did not provide enough background information on the H7N9 virus, which caused much more serious problems than H7N7. The H7N9 low pathogenic viruses emerged in China in 2013 mutated to highly pathogenic strains in 2017 (PMID: 23868922, PMID: 29151586), and studies have reported that the viruses were well controlled by the application of vaccine in poultry (PMID: 23868922, PMID: 30269969, PMID: 33905456, PMID: 30414008). A brief summary of these progress of H7N9 virus research and control in the introduction will make the manuscript much more informative.
- Page 4. Line3. “infectivity dose (EID50)/mL in a volume of 50 μL of 20X-20 (H7N9) and” I assume the mice were inoculated with 107 EID50 of the virus, so the “/mL” should be deleted.
- Page 4 line 12. the egg infectivity dose (EID50) should be changed to EID50, because the abbreviation has already appeared on page 4 line 3.
- Page 5. The amino acids at positions 190, 225, 226, 227, and 228 of the HA gene (H3 numbering) are related to host receptor binding efficiency. The key mutations for 2013 H7N9 are the amino acids at positions 186 and 226 (PMID: 32205415), therefore, the amino acids at 186 should be analyzed.
- Table 1. PB2. PB2 I292V was reported to increase the polymerase activity in both human and avian cells (PMID: 31597771). Please analyzed the amino acid at this position.
- Table 1. PB2 E627K and D701N rows: the “increased virulence in” and “mice” are separated in two rows, please correct. In addition. The citation for 627 and 701 is not suitable. Please cite proper papers highlighting the importance of PB2 627 (PMID: 8445709) and PB2 701 (PMID: 20041223 and PMID: 31213560).
- Table 1. NP. The V286A and M437T in NP increase the virulence of H7N9 virus in mice (PMID: 31666373). Please analyzed these sites as well.
- Figure 2: Please provide a higher resolution figure.
- Figure 3. Line 73. Please described the experiment correctly. Page 5. The amino acids at positions 190, 225, 226, 227, and 228 of the HA gene (H3 numbering) are related to host receptor binding efficiency. The key mutations for 2013 H7N9 are the amino acids at positions 186 and 226, therefore, the amino acids at 186 should be analyzed and a proper paper (PMID: 23787694) should be cited.
- BALB/c mice were intranasally inoculated with 107EID50 of each virus, not 107EID50/mL.
- Lines 143 to 157. The difference between the two isolates should be incorporated into results 3.1, and the potential risk caused by mutations of key amino acids should be discussed.
Author Response
Dear Yiska Xiong
Thank you for inviting us to submit a revised draft of our manuscript entitled, "Genetic Characterization and Pathogenicity of H7N7 and H7N9 Avian Influenza Viruses Isolated from South Korea" to Viruses. We also appreciate the time and effort you and each of the reviewers have dedicated to providing insightful feedback on ways to strengthen our paper. Thus, it is with great pleasure that we resubmit our article for further consideration. We have incorporated changes that reflect the detailed suggestions you have graciously provided. We also hope that our edits and the responses we provide below satisfactorily address all the issues and concerns you and the reviewers have noted.
To facilitate your review of our revisions, the following is a point-by-point response to the questions and comments.
The revised sentences were showed in blue.
# Reviewer 1
- Page 1. Abstract. The authors described that “There were no amino acid substitutions at the receptor-binding site of the HA”. The authors should indicate a reference strain that they compared with in their analysis. Without a reference virus, it’s difficult to understand the so-called substitutions
RESPONSE: We described A/Anhui/1/2013 (H7N9) as a reference strain. It was reported that A/Anhui/1/2013 virus is strongly interacted with α2,6-linked sialic acids receptor.
- Page 1. Introduction. Several important papers (PMID: 29151586 and PMID: 7483266) reporting the findings that the H5 and H7 viruses have the potential to mutate from LPAIV to HPAIV should be cited in the area where the reference 3 is cited.
RESPONSE: As you advised, we added other references.
- The authors did not provide enough background information on the H7N9 virus, which caused much more serious problems than H7N7. The H7N9 low pathogenic viruses emerged in China in 2013 mutated to highly pathogenic strains in 2017 (PMID: 23868922, PMID: 29151586), and studies have reported that the viruses were well controlled by the application of vaccine in poultry (PMID: 23868922, PMID: 30269969, PMID: 33905456, PMID: 30414008). A brief summary of these progress of H7N9 virus research and control in the introduction will make the manuscript much more informative.
RESPONSE: We added information related to H7N9 viruses in introduction as your kind recommendation.
- Page 4. Line3. “infectivity dose (EID50)/mL in a volume of 50 μL of 20X-20 (H7N9) and” I assume the mice were inoculated with 107EID50of the virus, so the “/mL” should be deleted.
RESPONSE: We deleted “/ml”
- Page 4 line 12. the egg infectivity dose (EID50) should be changed to EID50, because the abbreviation has already appeared on page 4 line 3.
RESPONSE: We changed “the egg infectivity dose (EID50)” to “EID50”
- Page 5. The amino acids at positions 190, 225, 226, 227, and 228 of the HA gene (H3 numbering) are related to host receptor binding efficiency. The key mutations for 2013 H7N9 are the amino acids at positions 186 and 226 (PMID: 32205415), therefore, the amino acids at 186 should be analyzed.
RESPONSE: The amino acid at position 186 of the HA gene was analyzed. The results showed that there was no amino acid mutation in HA gene of 20X-20 and 34X-2. The results also describes in table 1.
- Table 1. PB2. PB2 I292V was reported to increase the polymerase activity in both human and avian cells (PMID: 31597771). Please analyzed the amino acid at this position.
RESPONSE: We analyzed the amino acid at position 292 of PB2 gene. Except for 292V in Anhui/1, all H7 viruses including two H7 Korean AIVs had 292I. The results describes in table 1.
- Table 1. PB2 E627K and D701N rows: the “increased virulence in” and “mice” are separated in two rows, please correct. In addition. The citation for 627 and 701 is not suitable. Please cite proper papers highlighting the importance of PB2 627 (PMID: 8445709) and PB2 701 (PMID: 20041223 and PMID: 31213560).
RESPONSE: We corrected the sentence “increased virulence in mice” to fit in one compartment. Also, we changed references of PB2 627 and 701.
- Table 1. NP. The V286A and M437T in NP increase the virulence of H7N9 virus in mice (PMID: 31666373). Please analyzed these sites as well.
RESPONSE: We analyzed the amino acid at position V286A and M437T of NP gene. All H7 viruses showed amino acid substitution and these results described in Table 1
- Figure 2: Please provide a higher resolution figure.
RESPONSE: We changed Figure 2 with a higher resolution image.
- Figure 3. Line 73. Please described the experiment correctly. Page 5. The amino acids at positions 190, 225, 226, 227, and 228 of the HA gene (H3 numbering) are related to host receptor binding efficiency. The key mutations for 2013 H7N9 are the amino acids at positions 186 and 226, therefore, the amino acids at 186 should be analyzed and a proper paper (PMID: 23787694) should be cited.
RESPONSE: As I described in 12, we changed “107EID50/mL” to “107EID50”. Also, the latter comments is overlaps with number 6. We provided response in number 6.
- BALB/c mice were intranasally inoculated with 107EID50 of each virus, not 107EID50/mL
RESPONSE: We changed “107EID50/mL” to “107EID50”
- Lines 143 to 157. The difference between the two isolates should be incorporated into results 3.1, and the potential risk caused by mutations of key amino acids should be discussed.
RESPONSE: The part of third paragraph of discussion is incorporated into results 3.2, which described molecular characterization of H7 AIV isolates. Also, additional manuscript about amino acid mutation of two H7 Korean AIVs is described in the third paragraph of discussion.
Again, thank you for giving us the opportunity to strengthen our manuscript with your valuable comments and queries. We have worked hard to incorporate your feedback and hope that these revisions persuade you to accept our submission.

Reviewer 2 Report
Summary
The manuscript by Na et al entitled, “Genetic Characterization and Pathogenicity of H7N7 and H7N9 avian influenza viruses isolated from South Korea” is a study on two avian influenza viruses isolated from fecal samples. Phylogenetic analyses showed that the two Korean H7 AIVs are from the Eurasian lineage, except for the one isolate having an M segment from the American lineage. Examination of H7 sequences also showed that the viruses would most likely have an avian receptor binding specificity and have amino acid changes in NA that are associated with antiviral resistance. Pathogenicity testing in mice demonstrated that it is possible to infect mice. Significant lung lesions and weight loss were observed in mice infected with one of the isolates.
Overall comments and suggestions:
The data reported in the manuscript are valuable and are important contributions to our understanding of H7 avian influenza viruses that can cross the avian-mammalian species barrier. However, I would suggest a more careful presentation of the data and the conclusions that are drawn from the data points. Also, please carefully review references and make sure that the information cited are accurate.
Specific comments and suggestions:
Introduction:
- Line 37-39: H1-16 and N1-9 are avian in origin, but other HA and NA subtypes in bats have been discovered (H17-18, N10-11).
- Tong, Suxiang et al. “New world bats harbor diverse influenza A viruses.” PLoS pathogens vol. 9,10 (2013): e1003657. doi:10.1371/journal.ppat.1003657
- Tong S, Li Y, Rivailler P, et al. A distinct lineage of influenza A virus from bats. Proc Natl Acad Sci U S A. 2012;109(11):4269-4274. doi:10.1073/pnas.1116200109
- Line 42-44: Other examples are available for H7 outbreaks that lead to emergence of HPAIV from LPAIV
- Canada
- Berhane Y, Hisanaga T, Kehler H, Neufeld J, Manning L, Argue C, Handel K, Hooper-McGrevy K, Jonas M, Robinson J, Webster RG, Pasick J. 2009. Highly pathogenic avian influenza virus A (H7N3) in domestic poultry, Saskatchewan, Canada, 2007. Emerg Infect Dis 15:1492-5.
- Chile
- Suarez DL, Senne DA, Banks J, Brown IH, Essen SC, Lee C-W, Manvell RJ, Mathieu-Benson C, Moreno V, Pedersen JC, Panigrahy B, Rojas H, Spackman E, Alexander DJ. 2004. Recombination resulting in virulence shift in avian influenza outbreak, Chile. Emerging Infect Dis 10:693-699.
- Netherlands
- Beerens N, Heutink R, Harders F, Bossers A, Koch G, Peeters B. 2020. Emergence and Selection of a Highly Pathogenic Avian Influenza H7N3 Virus. J Virol 94:e01818-19.
- United States
- Lee DH, Torchetti MK, Killian ML, Berhane Y, Swayne DE. 2017. Highly Pathogenic Avian Influenza A(H7N9) Virus, Tennessee, USA, March 2017. Emerg Infect Dis 23.
- Killian ML, Kim-Torchetti M, Hines N, Yingst S, DeLiberto T, Lee DH. 2016. Outbreak of H7N8 Low Pathogenic Avian Influenza in Commercial Turkeys with Spontaneous Mutation to Highly Pathogenic Avian Influenza. Genome Announc 4.
- Youk S, Lee D-H, Killian M, Pantin-Jackwood M, Swayne D, Torchetti M. 2020. Highly Pathogenic Avian Influenza A(H7N3) Virus in Poultry, United States, 2020. Emerging Infectious Disease journal 26.
- United Kingdom
- Byrne AMP, Reid SM, Seekings AH, Núñez A, Obeso Prieto AB, Ridout S, Warren CJ, Puranik A, Ceeraz V, Essen S, Slomka MJ, Banks J, Brown IH, Brookes SM. 2021. H7N7 Avian Influenza Virus Mutation from Low to High Pathogenicity on a Layer Chicken Farm in the UK. Viruses 13.
- Line 46-48: Prevalence of avian influenza in wild birds have seasonality and predominant subtypes change every year. There is plenty of evidence for this. Please see below for an example. Thus, if reporting/citing prevalence, it is important to qualify the time and location where the survey was conducted.
- Dugan VG, Chen R, Spiro DJ, Sengamalay N, Zaborsky J, Ghedin E, Nolting J, Swayne DE, Runstadler JA, Happ GM, Senne DA, Wang R, Slemons RD, Holmes EC, Taubenberger JK. 2008. The evolutionary genetics and emergence of avian influenza viruses in wild birds. PLoS Pathog 4:e1000076.
- Line 60: “Infected women” to “infected woman”
- Line 60-62: The H7N7 cases in humans from reference 13 (Fouchier et al, 2004) is from the Netherlands and not from Italy.
- Canada
Methods:
- Line 90: “2,800 g” to “2,800 x g”
- Line 483-484: Please check reference 20. The author is listed as “Leptospirosis” and should be OIE.
- Line 127-131: Please provide a rationale for the choice of reference sequences that were downloaded from NCBI and GSAID. Please also note that GISAID also requires acknowledgement of contributors of the sequences used. It would be good practice to list reference sequences used from NCBI and acknowledge those as well.
Results:
- Line 188-189: “The hosts of the two H7 Korean AIVs were Anas platyrhynchos”.
- As I understand it, a more accurate statement would be: “Cytochrome C oxidase I gene sequencing reveled that fecal samples came from Anas platyrhynchos (mallards).” In my opinion, it is always good practice to state the actual data used to draw conclusions from.
- Were the sequences for 20X-20 and 34X-2 submitted to BLAST? If so, what were the results?
- Figure 2
- Please provide units for the x-axis.
- Please provide a higher resolution image. The strain names are barely readable.
- Please provide a rationale for having both Figure 1 and 2 in the manuscript. As it is currently presented, it appears that Figures 1 and 2 are redundant. The G1 and G2 groups can be marked in Figure 1 since the similar sequences were used (at least as far as I can guess since the strain names are unclear in Figure 2). The time-scaled tree in Figure 2 would be useful if timing and origin of the H7 isolates are of interest.
Discussion
- Line 352: There are two basic residues in the reported cleavage site: ELPKGR/GLF. It is thus not a monobasic cleavage site. This is types of unusual cleavage sites with 2-3 basic amino acid have been reported. The same cleavage site has been reported in human case (reference 9):
- Xiang Huo, Lun-biao Cui, Cong Chen, Dayan Wang, Xian Qi, Ming-hao Zhou, Xiling Guo, Fengming Wang, William J. Liu, Weirong Kong, Daxin Ni, Ying Chi, Yiyue Ge, Haodi Huang, Feifei Hu, Chao Li, Xiang Zhao, Ruiqi Ren and Feng-Cai Zhu. Severe human infection with a novel avian-origin influenza A(H7N4) virus. Science Bulletin 63, 1043 (2018); doi: 10.1016/j.scib.2018.07.003
- Line 354-357: The more accurate description is that amino acid substitutions with known effect on receptor binding to mammalian tissues were not found in the two Korean H7 AIVs, and thus the Korean H7 AIVs would likely have an avian receptor binding specificity. Since the characterization is only on the sequence level, actual binding experiments would need to be performed to demonstrate avian receptor binding specificity.
- Line 366: “antiviral effect” to “antiviral drug susceptibility”
- Line 379: “homology” to “identity”. Homology refers to whether two taxa are related or not and is a binary trait. In contrast, sequence identity refers to the degree of similarity between two sequences.
- Line 407-408: The phylogenetic analyses instead of the pathogenicity studies in mice provide better evidence for different gene constellations between 34X-2 and 20X-20. What the pathogenicity data in mice suggest is that potentially certain segments in 34X-2 may confer enhanced virulence, especially M, NP, and NA which are more distantly related to 20X-20 than other segments.
- Since three out of three lung tissues were positive for AIV at 1,3,5, and 7 dpi, the expectation would be that seroconversion occurred. Is it feasible to perform ELISA on the 14 dpi sera? It may be possible to pick up a low antibody levels with ELISA rather than HI. If not, can you explain lack of seroconversion in one mouse inoculated with 34X-2?
Author Response
Dear Yiska Xiong
Thank you for inviting us to submit a revised draft of our manuscript entitled, "Genetic Characterization and Pathogenicity of H7N7 and H7N9 Avian Influenza Viruses Isolated from South Korea" to Viruses. We also appreciate the time and effort you and each of the reviewers have dedicated to providing insightful feedback on ways to strengthen our paper. Thus, it is with great pleasure that we resubmit our article for further consideration. We have incorporated changes that reflect the detailed suggestions you have graciously provided. We also hope that our edits and the responses we provide below satisfactorily address all the issues and concerns you and the reviewers have noted.
To facilitate your review of our revisions, the following is a point-by-point response to the questions and comments.
The revised sentences were showed in Green.
# Reviewer 2
Introduction:
- Line 37-39: H1-16 and N1-9 are avian in origin, but other HA and NA subtypes in bats have been discovered (H17-18, N10-11).
Tong, Suxiang et al. “New world bats harbor diverse influenza A viruses.” PLoS pathogens vol. 9,10 (2013): e1003657. doi:10.1371/journal.ppat.1003657
Tong S, Li Y, Rivailler P, et al. A distinct lineage of influenza A virus from bats. Proc Natl Acad Sci U S A. 2012;109(11):4269-4274. doi:10.1073/pnas.1116200109
RESPONSE: As you mentioned, we revised manuscript and add references.
- Line 42-44: Other examples are available for H7 outbreaks that lead to emergence of HPAIV from LPAIV.
Canada; Berhane Y, Hisanaga T, Kehler H, Neufeld J, Manning L, Argue C, Handel K, Hooper-McGrevy K, Jonas M, Robinson J, Webster RG, Pasick J. 2009. Highly pathogenic avian influenza virus A (H7N3) in domestic poultry, Saskatchewan, Canada, 2007. Emerg Infect Dis 15:1492-5.
Chile; Suarez DL, Senne DA, Banks J, Brown IH, Essen SC, Lee C-W, Manvell RJ, Mathieu-Benson C, Moreno V, Pedersen JC, Panigrahy B, Rojas H, Spackman E, Alexander DJ. 2004. Recombination resulting in virulence shift in avian influenza outbreak, Chile. Emerging Infect Dis 10:693-699.
Netherlands; Beerens N, Heutink R, Harders F, Bossers A, Koch G, Peeters B. 2020. Emergence and Selection of a Highly Pathogenic Avian Influenza H7N3 Virus. J Virol 94:e01818-19.
United States; Lee DH, Torchetti MK, Killian ML, Berhane Y, Swayne DE. 2017. Highly Pathogenic Avian Influenza A(H7N9) Virus, Tennessee, USA, March 2017. Emerg Infect Dis 23.
Killian ML, Kim-Torchetti M, Hines N, Yingst S, DeLiberto T, Lee DH. 2016. Outbreak of H7N8 Low Pathogenic Avian Influenza in Commercial Turkeys with Spontaneous Mutation to Highly Pathogenic Avian Influenza. Genome Announc 4.
Youk S, Lee D-H, Killian M, Pantin-Jackwood M, Swayne D, Torchetti M. 2020. Highly Pathogenic Avian Influenza A(H7N3) Virus in Poultry, United States, 2020. Emerging Infectious Disease journal 26.
United Kingdom; Byrne AMP, Reid SM, Seekings AH, Núñez A, Obeso Prieto AB, Ridout S, Warren CJ, Puranik A, Ceeraz V, Essen S, Slomka MJ, Banks J, Brown IH, Brookes SM. 2021. H7N7 Avian Influenza Virus Mutation from Low to High Pathogenicity on a Layer Chicken Farm in the UK. Viruses 13.
RESPONSE: According to your kind advice, we added other references in manuscript.
- Line 46-48: Prevalence of avian influenza in wild birds have seasonality and predominant subtypes change every year. There is plenty of evidence for this. Please see below for an example. Thus, if reporting/citing prevalence, it is important to qualify the time and location where the survey was conducted.
Dugan VG, Chen R, Spiro DJ, Sengamalay N, Zaborsky J, Ghedin E, Nolting J, Swayne DE, Runstadler JA, Happ GM, Senne DA, Wang R, Slemons RD, Holmes EC, Taubenberger JK. 2008. The evolutionary genetics and emergence of avian influenza viruses in wild birds. PLoS Pathog 4:e1000076.
RESPONSE: According to the reference (PMID: 24423384), the research was conducted based on genome sequences and results of several surveillance systems. Since the survey was conducted without a limited time, it is difficult to describe exact time. However, it is clear that the H7N7 subtype is reported in the largest number of countries (21 countries) among the H7Nx subtype. The results are described in manuscript.
- Line 60: “Infected women” to “infected woman”
RESPONSE: We changed “Infected women” to “Infected woman”.
- Line 60-62: The H7N7 cases in humans from reference 13 (Fouchier et al, 2004) is from the Netherlands and not from Italy.
RESPONSE: We changed “Italy” to “Netherlands”.
Method
- Line 90: “2,800 g” to “2,800 x g”
RESPONSE: We changed “2,800 g” to “2,800 x g”.
- Line 483-484: Please check reference 20. The author is listed as “Leptospirosis” and should be OIE.
RESPONSE: We changed reference to “Stear, M. J. 2005. OIE Manual of Diagnostic Tests and Vaccines for Terrestrial Animals (Mammals, Birds and Bees) 5th Edition. Parasitology, Volume 130, Issue 6” which is updated version.
- Line 127-131: Please provide a rationale for the choice of reference sequences that were downloaded from NCBI and GSAID. Please also note that GISAID also requires acknowledgement of contributors of the sequences used. It would be good practice to list reference sequences used from NCBI and acknowledge those as well.
RESPONSE: In first paragraph of materials and methods 2.4, we described a rationale for the choice of reference sequences. In addition, we made supplementary materials table S1 to express our gratitude to the contributors of the reference sequences from GISAD and NCIB. As the reference sequence lists of Figure 2 in Supplementary materials table 1 is created based on GISAID accession number, we changed “GenBank” to “GISAID” in second paragraph of materials and methods 2.4.
Results
- Line 188-189: “The hosts of the two H7 Korean AIVs were Anas platyrhynchos”. As I understand it, a more accurate statement would be: “Cytochrome C oxidase I gene sequencing reveled that fecal samples came from Anas platyrhynchos (mallards).” In my opinion, it is always good practice to state the actual data used to draw conclusions from. Were the sequences for 20X-20 and 34X-2 submitted to BLAST? If so, what were the results?
RESPONSE: As you advised, we changed the manuscript. The sequences for 20X-20 and 34X-2 were deposited to Genbank and the accession numbers were described in line1-3 of Result 3.2.
Figure 2
- Please provide units for the x-axis.
RESPONSE: We describe what the horizontal axis and the unit indicates in Figure 2.
- Please provide a higher resolution image. The strain names are barely readable.
RESPONSE: We changed Figure 2 with a higher resolution image.
- Please provide a rationale for having both Figure 1 and 2 in the manuscript. As it is currently presented, it appears that Figures 1 and 2 are redundant. The G1 and G2 groups can be marked in Figure 1 since the similar sequences were used (at least as far as I can guess since the strain names are unclear in Figure 2). The time-scaled tree in Figure 2 would be useful if timing and origin of the H7 isolates are of interest.
RESPONSE: We provide rationales for Figure 2 in material and methods 2.2 and results 3.3. Phylogenetic tree of Figure 1 constructed using representative sequences of Eurasian lineages and American lineages without considering the time when the virus was isolated. However, phylogenetic tree of Figure 2 was considered the time and more reference sequences which includes only sequences isolated from Asia since 2016 were used than that of Figure 1. Therefore, it is difficult to distinct G1 and G2 lineage in Figure 1.
Discussion
- Line 352: There are two basic residues in the reported cleavage site: ELPKGR/GLF. It is thus not a monobasic cleavage site. This is types of unusual cleavage sites with 2-3 basic amino acid have been reported. The same cleavage site has been reported in human case (reference 9):
Xiang Huo, Lun-biao Cui, Cong Chen, Dayan Wang, Xian Qi, Ming-hao Zhou, Xiling Guo, Fengming Wang, William J. Liu, Weirong Kong, Daxin Ni, Ying Chi, Yiyue Ge, Haodi Huang, Feifei Hu, Chao Li, Xiang Zhao, Ruiqi Ren and Feng-Cai Zhu. Severe human infection with a novel avian-origin influenza A(H7N4) virus. Science Bulletin 63, 1043 (2018); doi: 10.1016/j.scib.2018.07.003
RESPONSE: We changed “monobasic amino acid” to “ELPKGR/GLF”. Also, I added the sentence “and the same cleavage site has been reported in human influenza virus”
- Line 354-357: The more accurate description is that amino acid substitutions with known effect on receptor binding to mammalian tissues were not found in the two Korean H7 AIVs, and thus the Korean H7 AIVs would likely have an avian receptor binding specificity. Since the characterization is only on the sequence level, actual binding experiments would need to be performed to demonstrate avian receptor binding specificity.
RESPONSE: We revised second paragraph of discussion. The sentence “There were no amino acid substitutions at the receptor-binding site of HA (E190D, Q/G225D, Q226L, S227N, and G228S), demonstrating that the two Korean H7 AIVs have avian receptor-binding specificity” changed to “There were no amino acid substitutions of G186V, E190D, Q/G225D, Q226L, S227N, and G228S in the HA gene with known effect on receptor binding to mammalian tissue, demonstrating that the two Korean H7 AIVs would likely have an avian receptor-binding specificity. However, since the characterization was analyzed only at the sequence level, actual receptor binding experiment would need to be performed to demonstrate avian receptor binding specificity”
- Line 366: “antiviral effect” to “antiviral drug susceptibility”
RESPONSE: We changed “antiviral effect” to “antiviral drug susceptibility”
- Line 379: “homology” to “identity”. Homology refers to whether two taxa are related or not and is a binary trait. In contrast, sequence identity refers to the degree of similarity between two sequences.
RESPONSE: As you pointed, we changed “homology” to “identity”
- Line 407-408: The phylogenetic analyses instead of the pathogenicity studies in mice provide better evidence for different gene constellations between 34X-2 and 20X-20. What the pathogenicity data in mice suggest is that potentially certain segments in 34X-2 may confer enhanced virulence, especially M, NP, and NA which are more distantly related to 20X-20 than other segments.
RESPONSE: We changed “This results suggests that the two viruses have different gene constellations” to “AIVs can increase virulence by reassortment with other AIVs and introduction of other segment may facilitate transmissions of AIVs (PMID: 21206092, PMID: 29669294). This result suggests that internal genes of 34X-2, especially M, NP and NA genes which clustered into distinct subgroups with 20X-20 may affect enhancing pathogenicity of 34X-2”
- Since three out of three lung tissues were positive for AIV at 1,3,5, and 7 dpi, the expectation would be that seroconversion occurred. Is it feasible to perform ELISA on the 14 dpi sera? It may be possible to pick up a low antibody levels with ELISA rather than HI. If not, can you explain lack of seroconversion in one mouse inoculated with 34X-2?
RESPONSE: As you suggested, we performed seroconversion test using commercial NP-ELISA kit (Bionote, Hwaseong, Korea). The ELISA kit is based on a competitive ELISA method and if the percent inhibition (PI) values are 50 or higher, the sample was considered positive (PI value = [1-(OD test sample/OD negative control)] x 100). The results of ELISA test showed the seroconversion was observed all of three mice infected with 34X-2 (PI value exceeded 50). However, ironically, two of three mice infected with 20X-20 showed negative result and one of three mice showed positive result. The PI value of two negative sample was 28.9 and 28.4, indicating slightly seroconversion occurred (average of negative PI value is 2.18). In addition, the ELISA kit recommended that the serum of chicken, duck, turkey, dog and pig can be examined. As the mouse serum may not appropriate to use the ELISA kit, the results of EILSA is not correspond to the HI results. According to (PMID: 29325601), the ELISA has better sensitivity than HI test, therefore the one of three mice infected with 34X-2 may occurred seroconversion but not enough to be detected with HI test.
Again, thank you for giving us the opportunity to strengthen our manuscript with your valuable comments and queries. We have worked hard to incorporate your feedback and hope that these revisions persuade you to accept our submission.
